# Presentation (Chopped Versus Ground and Pelleted) of a Low-Quality Alfalfa Hay in Sheep: Effects on Intake, Feeding Behaviour, Rumen Fill and Digestion, and Passage

**DOI:** 10.3390/ani15040541

**Published:** 2025-02-13

**Authors:** Antonio de Vega, Josep Gasa, Carlos Castrillo, José Antonio Guada

**Affiliations:** Departamento de Producción Animal y Ciencia de los Alimentos, Instituto Agroalimentario de Aragón (IA2), Universidad de Zaragoza-CITA, Miguel Servet 177, 50013 Zaragoza, Spain; josep.gasa@uab.cat (J.G.); ccastri@unizar.es (C.C.); jguada@unizar.es (J.A.G.)

**Keywords:** forages, sheep, particle size, intake, feeding behaviour, rumen fill, digestion, passage

## Abstract

Intake is the most important factor affecting animal performance, and in ruminants fed low-quality forages, the limiting factor for intake is rumen fill. Many factors affect rumen fill, among them the amount and rate of intake and the amount and rate of flow of digesta out of the rumen. The balance between those two factors will determine actual intake. Usually, animals fed forages ground and pelleted eat more than their counter partners offered the forages in the long form, and the question here is: Why? In the present work, the daily evolution of rumen fill in wethers fed low-quality alfalfa hay offered either chopped or ground and pelleted was studied. As factors influencing rumen fill, we assessed feeding behaviour (time spent chewing during eating and rumination), rumen fill (estimated by dilution of substances inserted in the rumen, known as markers), digestion in the rumen (incubating the feeds in polyester bags introduced in this compartment), and passage of the material out of the viscera (estimated with the aid of markers). The results showed that animals fed the pelleted hay ate more than those offered the long hay. As there was no effect of the diet on rumen volume or rate of digestion, a faster emptying of the liquid phase of the rumen, likely containing high amounts of small particles, was probably the reason.

## 1. Introduction

Intake is the most important factor affecting animal performance [1], and researchers have shown that in ruminants fed low-quality forages, the limiting factor for intake is rumen fill [2], which is in turn related to the rate and amount of intake and to the rate and amount of digesta flow out of the rumen [3,4,5]. This latter is influenced by comminution of particles (either physically, chewing during eating and ruminating [6,7,8], or chemically, rate and amount of degradation in the rumen [1]) and outflow rate of particles with a determined size [9,10]. Numerous experiments have been carried out to study the effect of the physical form of the forage on intake and rumen function, including digestion and passage [11,12,13,14,15], and the main reason for the higher voluntary intake of pelleted versus chopped low-quality forages seems to be a reduced retention time of digesta rather than changes in rumen fill [16].

Physical and chemical characteristics of dietary fibre affect mat formation (a pool of large particles not eligible for escaping the reticulorumen [1]) but also influence the ability of a feed to promote chewing and saliva secretion, which in turn affect ruminal fermentation and digesta passage [17,18]. The rate and extent of fibre digestion and the retention of different particle families in the reticulorumen depends on the interactions among different structures (e.g., carbohydrates, proteins, phenolic compounds, and minerals) of the complex cell wall matrix of forage plants and on its physical characteristics (e.g., buoyancy and comminuting properties of the forage particulate matter [1,19,20]).

Particle size is considered to be one of the most limiting factors (together with particle shape) of digesta passage out of the rumen [21]. Large particles inside the rumen, coming from masticated fragments and comminuting particles of different ages, are not eligible to escape the reticulorumen [1]. On the contrary, small comminuted and digested particles, resulting from a physicochemical breakdown that gradually depletes particles from their potentially digestible substrates and increases the proportion of indigestible ones, form part of the pool of fluid-diluted small particles that are eligible to escape the reticulorumen through the reticulo-omasal orifice [22,23,24,25]. It is worth mentioning that particles in the reticulorumen digesta are actually distributed in a wide continuous spectrum of particle sizes [26,27], and the flow of particles between the different pools is determined by both digestion and passage [24].

Even though the scientific community has collected a voluminous amount of information and knowledge about all the above-mentioned aspects, the mechanisms by which small particles in the rumen are selected for passage have not been clearly identified. On the one hand, small particles would be expected to behave as solutes [28,29], but on the other hand, there is entrapment of such particles in the rumen mat (pool of large particles not eligible for escaping the reticulorumen) [30]. Grinding and pelleting have been alleged to either enhance [30] or reduce [31] entrapment; hence, the relationships between particle size and digestion and passage kinetics are not sufficiently clear, as it is not the effect of rumen digesta particle size on marker dilution kinetics in the compartment.

The hypothesis of this work was that grinding and pelleting a low-quality alfalfa hay will increase its intake without altering rumen fill, hence modifying the outflow rate of digesta from the rumen. Also, we hypothesized that passage and digestion of small particles are the main factors affecting intake. On these grounds, the aim of the present experiment was to examine the behaviour of small particles within the rumen and determine if their retention time by passage or digestion pathways alter under-pelleting and control intake.

## 2. Materials and Methods

### 2.1. Animals and Diets

The experiment was carried out at the Servicio de Experimentación Animal (SEA) of the University of Zaragoza. Six four-year-old rumen-fistulated (5.3 cm i.d.) Rasa Aragonesa wethers, with an average live weight of 48.9 ± 1.58 kg, were allocated to one of two diets of alfalfa (*Medicago sativa*) hay, either chopped (50 mm, diet C) or ground (2 mm) and pelleted (diet P), in a crossover design. Animal care, handling, and surgical procedures were approved by the Ethics Committee of the University of Zaragoza (ethical approval code PI48/20). The care and management of animals were performed according to the Spanish Policy for Animal Protection RD 1201/05, which meets the EU Directive 86/609 on the protection of animals used for experimental and other scientific purposes. One month prior to the experiment, the animals were treated with albendazol (10 mL) to control internal parasites and kept in individual stands bedded on sawdust. Water and mineral blocks were available at all times throughout the experimental period.

### 2.2. Experimental Procedures

Alfalfa hay was offered ad libitum (seeking 20% refusals) once a day (9:00) as a unique ingredient, and the refusals were withdrawn 4 h later. The restriction in the access to the feed was imposed to allow for the study of rumen digesta kinetics and the effects of intake and chewing during eating and ruminating on rumen fill and digesta degradation in non-steady-state conditions. Each experimental period of the crossover lasted 25 d, of which the first 15 d were for adaptation to the treatments. During the following 20 d of each period, the amounts of hay offered to each animal were recorded daily. Representative samples of the hay were taken weekly throughout each experimental period, pooled, and analysed for chemical composition and particle size distribution (Table 1). Refusals were also collected daily and pooled per animal. Feeds and refusals were dried at 104 °C for 24 h to determine individual dry matter intake (DMI).

On day 16, the animals were given pulse doses of 25 g of Yb-labelled diets [32] and 50 mL of a solution containing 1.5 g of Co-EDTA [33] directly through the cannula before feeding for marker kinetic studies. Co-EDTA was added manually with the aid of a 50 mL syringe. Labelled diet C was previously chopped down with scissors to a size of about 5 mm to facilitate its introduction through the fistula. Labelled diets and Co-EDTA were deposited in four different places inside the rumen to allow for rapid homogenization. Samples of whole rumen content were taken at 1.5, 3, 6, 9, 12, 16, 24, 28, 36, 48, and 72 h after markers’ administration, and faeces samples were obtained directly from the rectum at 1.5, 3, 6, 9, 12, 16, 24, 28, 36, 48, 72, and 96 h. Subsamples of faeces and rumen content were immediately dried at 104 °C for 24 h, ground through a 1 mm sieve, and stored at room temperature until marker analysis. Another set of faeces subsamples was frozen at −20 °C until determination of particle size distribution, which was assessed on the pooled subsamples from all sampling times. On day 20, additional rumen samples (ca. 50 g) were obtained at 0 (before feeding), 4, 8, 12, 16, 20, and 24 h and were frozen at −20 °C until determination of DM and particle size distribution.

On day 21, particles of 0.5–1 mm length, 1–2 mm length, 2–4 mm length, and 4–8 mm length were incubated in polyester bags (10 × 16 cm; 48 µm pore size) in the rumen of the wethers for 3, 6, 10, 24, 48, and 72 h after feeding. The objective of this procedure was to assess the influence of the rumen environment set by the form of presentation of the alfalfa hay on rumen degradation of different particle sizes. Rumen degradability of dry matter (DM) was assessed following the procedures previously described [34]. About 5 g of particles of a determined size were incubated in triplicate bags per incubation time and wether. Particles were obtained by grinding samples of diet C (one per particle size to be obtained) through a sieve of the larger size aimed (1, 2, 4, and 8 mm for 0.5–1, 1–2, 2–4, and 4-8 mm particles, respectively). The ground material was then dry-sieved through the larger size for each case, and the material collected in the smaller size (0.5, 1, 2, and 4 mm for 0.5–1, 1–2, 2–4, and 4–8 mm particles, respectively) stored until incubation.

On day 24, 1.5 g of Co-EDTA diluted in 50 mm distilled water were introduced through the rumen cannula as described above, and rumen liquid samples (about 200 mL) were taken at 1, 1.5, 2, 3.5, 5, 8, 11, 14, 18, and 24 h. Rumen fluid was removed using a customized vacuum device connected to a 0.6 cm i.d. semi-rigid tube with 2 mm pores. Representative samples were taken, moving the tube in all directions inside the rumen while sampling. Samples were filtered through four layers of gauze and then stored at −20 °C until analysis of Co-EDTA. The purpose of this procedure was to study the dilution rate of the liquid phase of the ruminal digesta and the amount of liquid volume inside the rumen over a period of 24 h. Feeding behaviour was assessed on the next day, as previously described [35]. Visual observations were made once every 3 min on each animal, and three categories were used to classify the animals’ behaviour: chewing during eating, ruminating, and idling. Animals were weighed at the beginning and the end of each experimental period.

### 2.3. Chemical Analysis and Particle Size Distribution

Samples of the two diets were ground in a hammer mill (Retsch SM100 Comfort, Haan, Germany) fitted with a 1 mm sieve size and analysed following the procedures of AOAC [36] for DM (ref. 934.01), organic matter (OM; ref. 942.05), crude protein (CP; ref. 976.05), and ether extract (EE; ref. 2003.05). Concentration of neutral detergent fibre (aNDFom) was analysed as described by Mertens [37], using α-amylase and sodium sulphite, and results were expressed exclusive of residual ashes. Acid detergent fibre (ADFom) and acid detergent lignin (Lignin (sa)) were analysed as described by AOAC [36] (ref. 973.18) and Robertson and Van Soest [38], respectively. Particle size distribution was assessed as described previously [35].

Concentration of Co and Yb in rumen content [32] and faeces [35] was carried out at the Servicio de Análisis Químicos of the University of Zaragoza by atomic emission spectrometry, using a Perkin-Elmer P-40 spectrophotometer (Perkin-Elmer, Uberlingen, Germany). For Co determination in rumen liquid, samples were thawed and centrifuged at 2500× *g* for 10 min. Analysis was performed in 25 mL of the supernatant. Calibration standards were made using blank rumen contents or faeces, as appropriate. Particle size distribution of faeces and rumen content was determined using a wet-sieving apparatus, as previously described [32].

### 2.4. Mathematical and Statistical Methods

Faecal marker excretion curves were fitted to a multi-compartmental model [39], which was restricted to two compartments, as the rumen and caecum are assumed to be the main functional mixing pools of the gut. Markers’ fractional outflow rate from the rumen, estimated from dilution curves in this compartment, was calculated by linear transformation of a first-order kinetic [40]. The slope of the linear regression of logarithmic concentration of the markers (DM basis) on time was considered to represent the fractional outflow rate from the rumen. The use of markers is based on the assumption that rumen volume remains constant and that inflows to and outflows from the compartment are homogeneous, continuous, and equivalent [9,41], although it is known that these ideal conditions are rarely accomplished. In the present experiment, steady-state conditions were not met, but the pattern of marker excretion was generally repeated over 24 h periods. In these conditions, fractional outflow rate of Yb and Co from the rumen was estimated using only concentrations separated by 24 h periods (24, 48, and 72 h). Faecal output was estimated from faecal Yb and Co excretion curves [42] and used, together with intake, to estimate digestibility.

The disappearance of DM from polyester bags on time was fitted to the following first-order kinetic equation [43]: y = a + b (1 − e^−ct^), where “y” represents degradation at a given time of incubation (“t”), “a” the soluble fraction in the rumen, “b” the non-soluble but potentially degradable fraction, and “c” the fractional rate of degradation of fraction “b.” Potential degradability is represented by the sum of “a” and “b”.

Regarding the dilution rate of Co-EDTA in the rumen liquid, it was observed (Figure 1) that fitting the data to two regressions resulted in higher values of r^2^ for each individual regression. On these grounds, two Co dilution rates were obtained for each animal and period: the fast dilution rate (k_f_) was obtained, fitting the logarithms of concentrations that decayed abruptly to linear regression and slow dilution rate (k_s_) by fitting the rest of the points.

In the present work, it was assumed thatk_f_ = k_m_ + k_v_
being k_m_ the average daily dilution rate and k_v_ the rate of liquid-volume increase and thatk_s_ = k_m_ − k_v’_
being k_v’_ the rate of liquid-volume decrease. For a period of 24 h,k_v_ * T_fs_ = k_v’_ * (24 − T_fs_), andk_m_ = k_f_ * (T_fs_/24) + k_s_ * (1 − T_fs_/24),
being T_fs_ the time at which the regressions for k_f_ and k_s_ coincide. In a period of 24 h, k_m_ can be calculated as the weighed mean of k_f_ and k_s_.

The minimum daily volume of liquid in the rumen (Lmin) was obtained from the intercept of k_f_. Knowing the amount of marker dosed (Q) and the concentration in rumen liquid at time 0 (antilogarithm of the intercept, C_0_), thenLmin = Q/C_0_

The minimum daily content of whole digesta (WDmin) and DM (DMmin) were obtained from Lmin and the percentage of DM at 0 h in the samples for particle size distribution analysis.

The maximum daily volume of liquid in the rumen was estimated as [44]Lmax = Lmin + Lmin * (k_f_ − k_m_) * T_fs_, orLmax = Lmin + Lmin * (k_m_ − k_s_) * (24 − T_fs_)

The maximum daily content of whole digesta (WDmax) and DM (DMmax) were obtained from Lmax and the percentage of DM at 4 h in the samples for particle size distribution analysis.

Knowing DMmin and DMmax and the particle size distribution of rumen DM before (0 h) and 4 h after feeding, the amount of particles with a size >1.2 mm was estimated for these sampling times. Subsequently, the amount of particles that reduced their size to <1.2 mm (y) during eating (4 h) was calculated asy (g) = Intake of particles > 1.2 mm (g) + Particles > 1.2 mm in the rumen (g) at 0 h − Particles > 1.2 mm in the rumen (g) at 4 h

Similarly, the amount of particles that reduced their size to <1.2 mm (z) during rumination (between 4 and 24 h after feeding) was calculated asz (g) = Particles > 1.2 mm in the rumen (g) at 4 h − Particles > 1.2 mm in the rumen (g) at 24 h

Knowing the time dedicated to eating and ruminating, the rate of particle size reduction to a size <of 1.2 mm (g/h) during both activities can be calculated. Also, the proportion of feed particles that have been reduced during eating (y/Intake of particles > 1.2 mm (g)) and ruminating (z/Intake of particles > 1.2 mm (g)) can be obtained.

Data were analysed using the PROC MIXED of SAS (SAS Inst. Inc., Cary NC, USA, v 9.4). The fixed factors included in the ANOVA were treatment (C or P), period of the crossover, and their interaction. Animal was used as a random factor. Marker kinetic studies also included sampling site (rumen or faces) and interactions treatment * sampling site, period * sampling site and treatment * period * sampling site as fixed factors. As the period effect was not significant, a second ANOVA was performed, including fixed effects of marker (Co or Yb), treatment, sampling site, and all their interactions. In situ degradation studies included, as fixed effects, the particle size incubated (0.5–1 mm, 1–2 mm, 2–4 mm, or 4–8 mm) and interactions treatment * particle size, period * particle size and treatment * period * particle size. For all data, differences were considered significant if *p* < 0.05.

## 3. Results

DMI was higher (*p* < 0.001) in animals fed P than C (Table 2). The form of the diet consumed also influenced (*p* < 0.001) the intake of the different particle sizes, and animals fed C ingested more particles of a size >1.2 mm but less particles of a size between 0.15 and 1.2 mm or <0.15 mm (Table 2). The period effect and the interaction between treatment and period were not significant (*p* > 0.05) in any case. Estimated faecal output did not differ (*p* > 0.05) between markers (diet C: 599 g/day vs. 589 g/day for Yb and Co, respectively; diet P: 876 g/day vs. 863 g/day for Yb and Co). Digestibility values shown in Table 2 are those obtained using faecal output estimated from faecal excretion curves of Yb and were lower in animals fed diet P than C.

Animals fed diet C spent more time eating compared to animals fed diet P (Figure 2 and Table 3), expressed either as min/d or as min/kg DMI. Time destined to ruminating was also higher with diet C, but the differences with diet P were significant only when the time was expressed as min/kg DMI. Rumination did not take place during the eating period for any of the diets. Between 4 and 8 h post-feeding, rumination took longer time in animals fed diet C than P (Figure 2), whereas between 8 and 24 h post-feeding, the opposite happened.

Figure 3 shows the particle size distribution of rumen content and faeces; Table 3 also includes the rate of particle size reduction to a size <1.2 mm (g/h) during eating and ruminating and the proportion of feed particles that have been reduced during both activities. The proportion of particles >1.2 mm in faeces was negligible (<1.1%) with both diets. With respect to rumen content, the proportion of particles <0.15 mm was similar with both diets (average daily values of 39.8% vs. 42.4% for diets C and P, respectively). Regardless of the diet, there was an increase in the proportion of particles <0.15 mm 4 h after feeding, a constant proportion for the next 12 h, and a reduction 8 h before next feeding. Diet C promoted a higher proportion of particles >1.2 mm in the rumen than diet P (average daily values of 18.1% and 4.0%, respectively). In animals consuming the chopped hay, there was an increase of 58% of these particles 8 h after feeding and then a decrease until 24 h. In animals fed diet P, on the other hand, the proportions were nearly constant until 8 h, with an 86% increase between 8 h and 20 h, and then a 69% decrease in the last 4 h before next-day feeding.

Table 4 shows k_f_, k_s_, T_fs_, k_m_, and estimated minimum and maximum volumes of liquid, whole digesta, and DM inside the rumen of sheep fed C or P diets. As pointed out above (Figure 1), fitting the data to two regressions resulted in higher values of r^2^ for each individual regression (0.979 vs. 0.858 for k_f_ vs. a unique regression and 0.954 vs. 0.858 for k_s_ vs. a unique regression; *p* < 0.01. Differences between determination coefficients for k_f_ and k_s_ (0.979 vs. 0.954) were not significant).

All fractional dilution rates (k_f_, k_s_, and k_m_) were higher in animals fed P than C (*p* < 0.05). Regardless of the form of the hay, k_f_ was about 5.5 times higher than k_s_. The T_fs_ was not affected by the diet consumed and had an average value of 3.58 h. Due to the method of calculation, this time coincided with that when the maximum rumen volume was reached. Contrary to dilution rates, estimations of minimum rumen volume of liquid and WD were higher for diet P. However, there were no differences (*p* > 0.1) between diets in minimum volume of DM or in maximum volumes of liquid, WD, and DM. The DM proportion was higher in the rumen of animals consuming diet P (average daily values of 13.9% and 15.2% for diets C and P, respectively). Minimum values corresponded to the time before feeding and maximum values to 4 h (end of accession to the feed).

Fractional outflow rates of Co-EDTA and Yb-labelled particles from the rumen, estimated by either rumen or faeces sampling, are shown in Table 5. None of the factors considered (marker, treatment, or sampling site) had a significant effect (*p* > 0.1).

Potential degradability and fractional rate of degradation of the DM of the different particle sizes incubated in situ are shown in Table 6. These results only reflect the influence of the rumen environment as the particles incubated were the same in animals fed both C and P diets. The experimental period only affected potential degradability of particles with size within 1 and 2 mm (55.7% vs. 58.5% for periods 1 and 2, respectively), and as a result, the interaction between particle size and experimental period was significant (*p* = 0.03). On the other hand, differences between particle sizes were significant only in the second experimental period (56.3%b, 58.5%a, 55.3%b, and 53.2%c for particles of 0.5–1 mm, 1–2 mm, 2–4 mm, and 4–8 mm, respectively). Fractional rate of degradation was affected by particle size (0.114 h^−1^a, 0.101 h^−1^ab, 0.082 h^−1^c, and 0.085 h^−1^bc for particles with a size of 0.5–1 mm, 1–2 mm, 2–4 mm, and 4–8 mm, respectively), treatment (0.106 h^−1^ and 0.085 h^−1^ for C vs. P, respectively), and experimental period (0.102 h^−1^ and 0.089 h^−1^ for P1 and P2).

## 4. Discussion

Sheep fed diet C spent eating 90.4% of the time available for feed consumption (240 min), whereas in animals fed diet P, this figure went down to 39.4% (Table 3). These latter animals consumed 34% more feed (Table 2) in less than half the time, compared to animals fed diet C. This might indicate that intake was limited by rumen fill, which was reached in a short period of time for animals fed the P diet, as already discussed half a century ago [16,45,46] and more recently confirmed [47]. Digestibility values must be considered with precaution for several reasons: first of all, they were estimated from faecal output calculated using Yb faecal excretion curves. Lack of differences between faecal output estimated from Yb or Co faecal excretion curves has already been found [42], and in our case, Yb was chosen as Co is expected to be more associated with the liquid than with the solid phase of the digesta [33]. France and co-workers [42], however, concluded that both Yb and Co overestimated faecal output and hence underestimated digestibility, compared to measured values. On the other hand, the method for faecal output estimation is valid for steady-state conditions [42], and as pointed out above, ours were far from those. Despite all the previous considerations, however, estimated digestibility values were lower in animals fed P than C and in accordance with those previously found in sheep consuming low-quality alfalfa hay [35].

The estimated maximum amount of digesta in the rumen (Table 4) did not differ among diets, which could indicate that rumen distension was the limiting factor for intake in both C and P animals. At first sight, estimated volumes shown in Table 4 should be taken with care; minimum volumes were estimated from the intercept of k_f_ (see Section 2.4), and slight changes in this rate could result in important differences in estimated rumen volumes. On the other hand, maximum volumes were estimated using a method developed for dairy cows fed grass silage and concentrate twice a day ad libitum [44]. Moreover, obtaining a representative rumen sample for DM determination was not easy, and this might have influenced the estimations of WD and DM volumes.

In this study, the dilution rate of Co-EDTA in a period of 24 h was partitioned into a fast and a slow dilution rate (k_f_ and k_s_). Average intakes of 1212 g/day and 1622 g/day for diets C and P, respectively (Table 2), increased the whole digesta content of the rumen by 72% and 99% 4 h after feeding (Table 4). This increase in rumen volume is thought to be the main factor affecting the quick dilution of the marker in the compartment, together with the loss of Co-EDTA by passage (Table 5). The higher increment in rumen volume due to intake in animals fed diet P compared to C seems to be responsible for the higher k_f_ values, as there were no statistical differences in dilution rates between C and P (Table 5). The average time of intersection between k_f_ and k_s_ was 3.56 h, very close to the time where the hay was withdrawn after the feed offer. This would imply that k_s_ is due only to Co passage from the rumen, as there are no intake effects on its dilution. Higher values for the P than for the C diet agree with the well-known effect of intake on rate of passage [48]. To our knowledge, this is the first time k_f_ and k_s_ have been identified separately.

The estimated minimum volume of WD in the rumen represented 13.4% and 10.6% of the LW of the animals consuming diets C and P, respectively. Similar values were found in classical papers reporting data obtained by rumen emptying of animals fed leguminous hays offered once a day. In this respect, rumen volumes of 14.5% LW in ewes fed white clover hay at a rate of 50 g DM/kg LW^0.75^, lower than that reported in the present experiment (Table 2), have been found [49]. Also, values of 11.5% LW when feeding rams alfalfa hay at a rate of 64 g DM/kg LW^0.75^ have been obtained [50]. When grass instead of leguminous hays was offered, the minimum volume of WD in the rumen reached 20% of the LW [51]. The lack of differences in minimum rumen volume of DM between C and P might be a consequence of the higher DM content in the rumen of sheep fed diet P (average daily values of 13.9 and 15.2 g/100 g rumen content for diets C and P, respectively; *p* < 0.01).

In the present work, estimated maximum volumes of WD in the rumen also seem overestimated, reaching values of 23.1% and 21.2% of the LW for C and P. However, similar values have been previously obtained when feeding sheep either white clover hay (21.0% LW [49]) or ryegrass hay (23.9% LW [52]) at a rate of 50 g DM/kg LW^0.75^. As a consequence of the higher WDmin in animals fed diet C compared to P and the lack of differences in WDmax, rumen emptying occurred at a slower rate in the former animals. This is confirmed by the faster dilution rates of the liquid phase observed with diet P (Table 4).

Rate of passage of Co-EDTA and Yb-labelled particles (Table 5) did not differ between treatments, and this contradicts the last statement of the previous paragraph and the long-established idea of a faster rate for ground and pelleted diets [48,52,53], especially when the higher intake of particles <0.15 mm by animals fed diet P (Table 2) is compatible with a faster rumen emptying. However, it must be taken into account that marker dilution rates from the rumen were estimated with only three points (see Section 2.4), and this fact could have biased the results. The lack of differences between dilution rates of Co-EDTA and Yb-labelled particles could also be attributed, at least in part, to the method of calculation. It seems that, in the non-steady-state conditions of the present experiment, the estimation of marker dilution rates in the rumen over three days leads to inaccurate values, even with the precautions stated in Section 2.4. Instead, k_s_ could represent more accurately the outflow rate of the liquid phase of the digesta from the rumen.

Regardless of the diet, the proportion of particles <1.2 mm in the faeces was negligible (less than 1%), which supports the idea of a critical size [54] for the particles to be able to leave the rumen. The estimated rate of reduction of feed particles to a size <of 1.2 mm during eating (Table 3) was higher with diet P. It must be taken into account that 95% of particles >1.2 mm in diet C were of a size >2.4 mm, whereas in diet P, this figure reached only 9%. Animals consuming diet C produced particles of a size between 1.2 and 2.4 mm and, at the same time, reduced the size of particles >1.2 mm but <2.4 mm. However, animals in diet P mainly carried out this latter activity. The same reasoning can be applied to the proportion of feed particles that reduced their size to <1.2 mm during eating.

Even though animals consuming diet C spent more time eating than animals consuming diet P (Table 3 and Figure 2), either as min/d or as min/kg DMI, there were differences in the time destined to rumination only when expressed per kg DMI, as a result of the higher intake in animals fed P. The estimated rate of reduction of feed particles to a size <of 1.2 mm during rumination was also higher with diet C, which is in accordance with the higher amount of particles >1.2 mm in the rumen of the animals consuming the chopped forage (Figure 3).

Contrary to our results, other authors [30,55,56] have shown a higher reduction of feed particle size during ruminating than during eating. However, these authors refer to the % of the DM that is reduced to a size <of 1.2 mm, not to the proportion of particles >1.2 mm that are reduced. On the other hand, most authors collected their samples from the cardias and not directly from the rumen. When our results were expressed as the proportion of ingested DM that was reduced to a size <of 1.2 mm during eating, rather than to the proportion of particles >1.2 mm, the figures went down to 47.7% and 28.1% for diets C and P, respectively. These values were comparable to those found in the literature. Regarding rumination, values found in the literature refer to the relationship between the proportions of particles >1.2 mm in the regurgitated bolus before and after rumination so they cannot be compared to our results.

When the effectiveness in reducing diet particle size of chewing during eating and rumination was considered jointly, 91.6% and 97.4% of particles >1.2 mm were reduced for diets C and P, respectively. These values are rather high compared to those found in the literature, but it must be taken into account that in the present work, the samples were obtained directly from the rumen and that some reduction of size might have been blamed on microbial action or friction of the particles with the rumen wall [19].

Results shown in Table 6 only represent the influence of the rumen environment on rumen degradation of different particle sizes from alfalfa hay. Even though potential degradability was affected by the interaction between particle size and experimental period, differences were small in quantitative terms and hence considered not important. Regarding the fractional rate of degradation, it decreased when increasing the particle size, probably due to the fact of a decreased relationship surface/volume for the latter, and hence a decreased microbial attachment and activity per volume unit. With respect to the lower “c” values found for diet P, we can speculate about the more difficult access to the bags of the microbes in a visually observed more viscous environment. It must be pointed out that these results would not be comparable to the degradation values of the raw diets because of the method of obtaining the different particle sizes. In addition, if we accept that the particle size reduction is a time-dependent function ([57], the smaller the particle size would mean a longer time subjected to digestion. Then, actual fractional rate of degradation of a given particle size would be slower than that obtained with the in situ technique in the present experiment. Overestimations would increase with decreasing particle size.

## 5. Conclusions

Higher intake of ground (2 mm) and pelleted alfalfa hay compared to chopped (50 mm) alfalfa hay cannot be explained by higher maximum rumen volumes or faster degradation rates. Instead, the faster dilution rate of the liquid phase, likely including significant amounts of particles <0.15 mm, would be the main factor responsible.

This article is a revised and expanded version of a paper entitled “Revisiting particle kinetics in the rumen: comminution, digestion and passage functions as affected by diet type,” which was presented at the Joint Seminar of the FAO-CIHEAM Subnetworks on Production Systems and on Nutrition, of the FAO-CIHEAM Network on Sheep and Goats, held in Vitoria-Gasteiz, Spain, on 3–5 October 2017 [58].

## Figures and Tables

**Figure 1 animals-15-00541-f001:**
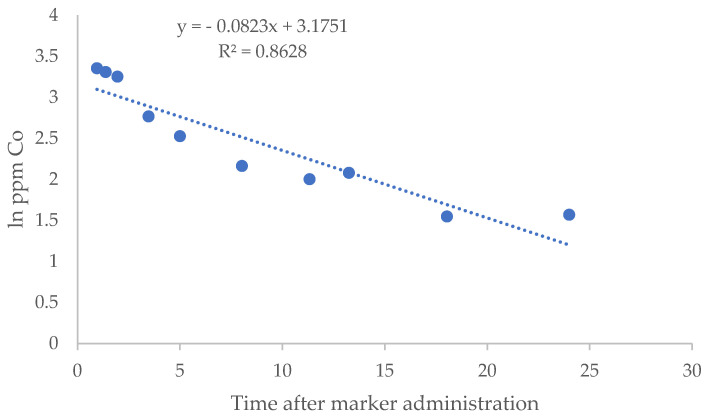
Example of the evolution of the logarithm of Co concentration in the rumen liquid of one animal for a period of 24 h. Fitting a unique regression on time had a low r^2^ value. When fitting two regressions (one with the first five points, in this example, and the other with the second five points), the r^2^ values improved substantially.

**Figure 2 animals-15-00541-f002:**
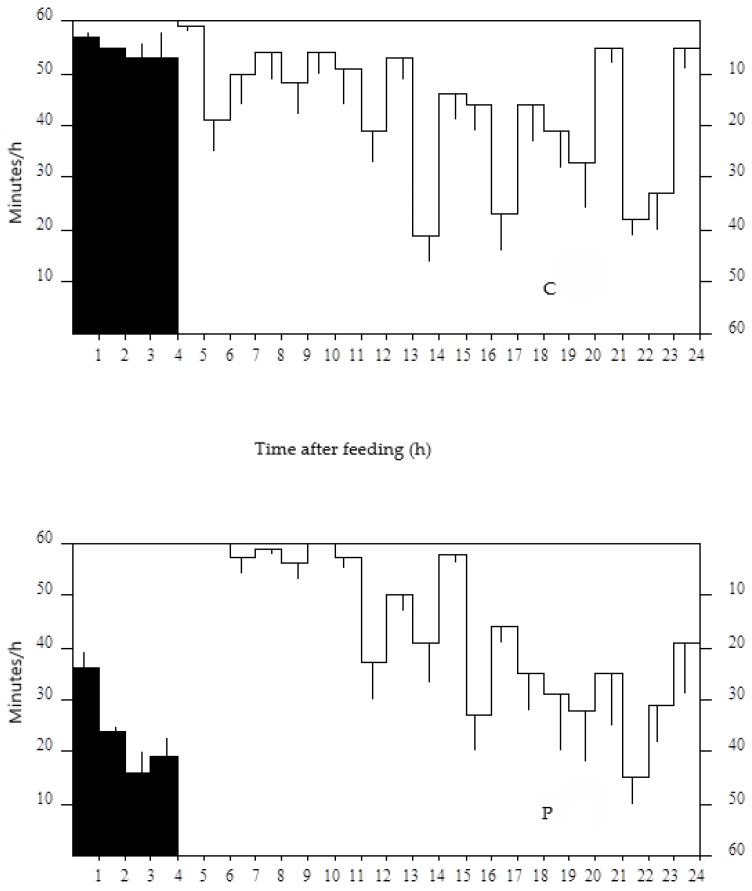
Daily pattern of eating (■) and rumination (□) for sheep fed chopped (50 mm, C) or ground (2 mm) and pelleted (P) alfalfa hay. Accession to the forage was restricted to 4 h. Bars represent the standard error.

**Figure 3 animals-15-00541-f003:**
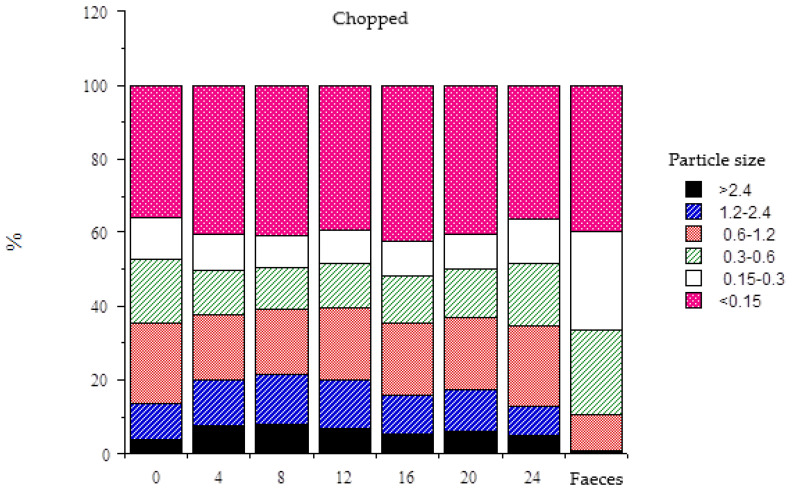
Percentages of different particle sizes (mm) in the rumen content at different times after feeding and in faeces of sheep fed chopped (50 mm, C) or ground (2 mm) and pelleted (P) alfalfa hay. Access to the forage was restricted to 4 h.

**Table 1 animals-15-00541-t001:** Nutrient composition and particle size distribution of alfalfa hay (Medicago sativa) chopped (50 mm, diet C) or ground (2 mm) and pelleted (diet P).

Treatment	C	P
Dry matter (DM, g/kg)	847	877
Organic matter (g/kg DM)	857	832
Crude protein (g/kg DM)	197	182
Ether extract (g/kg DM)	22	24
Neutral detergent fibre (g/kg DM)	431	457
Acid detergent fibre (g/kg DM)	312	315
Lignin (g/kg DM)	75	89
Particles > 1.2 mm (g/kg DM)	703	301
Particles > 0.15 mm and < 1.2 mm (g/kg DM)	32	357
Particles < 0.15 mm (g/kg DM)	265	342

**Table 2 animals-15-00541-t002:** Intake of DM and different particle sizes of alfalfa hay chopped (50 mm, C) or ground (2 mm) and pelleted (P) and digestibility values (g/100 g, estimated from faecal marker excretion curves of Yb) of dry matter (DMD), organic matter (OMD), crude protein (CPD), and neutral detergent fibre (NDF) in wethers.

Treatment	C	P	SEM	*p*-Value
Dry matter intake				
(g/day)	1212	1622	68.7	0.0010
(g/kg LW^0.75^)	65.4	87.1	3.59	0.0009
Intake of particles > 1.2 mm				
(g/day)	851	488	37.1	<0.0001
(g/kg LW^0.75^)	45.9	26.2	1.01	<0.0001
Intake of particles < 1.2 mm and > 0.15 mm				
(g/day)	38.6	579	17.27	<0.0001
(g/kg LW^0.75^)	2.09	31.1	1.165	<0.0001
Intake of particles < 0.15 mm				
(g/day)	322	555	21.1	<0.0001
(g/kg LW^0.75^)	17.4	29.8	1.18	<0.0001
DMD	52.4	47.7	0.49	<0.001
OMD	53.2	48.3	0.53	<0.001
CPD	68.7	59.9	0.55	<0.001
NDFD	38.1	31.6	1.22	<0.01

SEM: standard error of the mean; LW: live weight.

**Table 3 animals-15-00541-t003:** Feeding behaviour, rate of reduction of feed particles to a size <of 1.2 mm (g/h), and proportion of feed particles reduced to a size <of 1.2 mm (g/100 g) during eating and ruminating of sheep fed alfalfa hay chopped (50 mm, C) or ground (2 mm) and pelleted (P). Accession to the forage was restricted to 4 h. The period of the crossover effect and the interaction between treatment and period were not significant (*p* > 0.05) in any case.

Treatment	C	P	SEM	*p*-Value
E (min/day)	217	94.6	13.03	<0.0001
E (min/kg DMI)	180	58.2	11.85	<0.0001
Rate of reduction of feed particles during eating (g/h)	159	294	17.7	<0.0001
Percentage of feed particles reduced during eating	67.8	91.4	1.67	<0.0001
R (min/day)	350	275	42.4	0.1290
R (min/kg DMI)	287	170	28.5	0.0063
Rate of reduction of feed particles during rumination (g/h)	48.6	9.43	2.992	<0.0001
Percentage of feed particles reduced during rumination	23.8	6.02	1.525	<0.0001

E: eating; R: ruminating; DMI: dry matter intake; SEM: standard error of the mean.

**Table 4 animals-15-00541-t004:** Fast (k_f_; h^−1^) and slow (k_s_; h^−1^) dilution rates of the liquid phase of the rumen digesta, time of intersection between the two dilution rates (T_fs_; h), weighed average dilution rate of the liquid phase of rumen digesta (k_m_; h^−1^, see Section 2.4), and minimum and maximum volumes of liquid (Lmin and Lmax; litres, see Section 2.4), whole digesta (WDmin and WDmax; g, see Section 2.4), and dry matter (DMmin and DMmax; g, see Section 2.4) inside the rumen of sheep fed alfalfa hay chopped (50 mm, C) or ground (2 mm) and pelleted (P). Access to the feed was restricted to 4 h.

Treatment (T)	C	P	SEM	*p*-Value		
				T	Period (P)	T*P
k_f_	0.269	0.350	0.0236	0.0141	0.7717	0.4440
k_s_	0.049	0.065	0.0045	0.0490	0.4524	0.9208
T_fs_	3.41	3.74	0.376	0.4097	0.5417	0.9898
k_m_	0.079	0.109	0.0038	0.0015	0.6803	0.6803
Lmin	5.97	4.76	0.388	0.0203	0.2131	0.1600
Lmax	9.71	8.79	0.615	0.1808	0.3699	0.3282
WDmin	6730	5405	428.9	0.0214	0.2369	0.1696
WDmax	11557	10738	677.5	0.2718	0.3012	0.3114
DMmin	755	648	70.0	0.1769	0.7418	0.5159
DMmax	1951	1841	98.2	0.3061	0.1357	0.3598

**Table 5 animals-15-00541-t005:** Dilution rate of Co-EDTA and Yb-labelled particles (k_r_; h^−1^) from the rumen of wethers fed lucerne hay either chopped at 5 cm (C) or ground (2 mm) and pelleted (P), and slope of the faecal excretion curve of the markers (k_fa_; h^−1^). Access to the feed was restricted to 4 h. All interactions were not significant (*p* > 0.05).

Treatment (T)	C	P	SEM		*p*-Value	
				Marker	T	Sampling Site ^1^
Co-EDTA						
k_r_	0.0468	0.0448	0.00531	0.3260	0.8960	0.9256
k_fa_	0.0526	0.0496				
Yb-labelled particles						
k_r_	0.0466	0.0498				
k_fa_	0.0418	0.0450				

SEM: standard error of the mean. ^1^ Rumen or faeces.

**Table 6 animals-15-00541-t006:** Potential degradability (a + b; g/100 g DM) and fractional rate of degradation (c; h^−1^) of DM of different particle size populations in the rumen of wethers fed lucerne hay either chopped at 5 cm (C) or ground (2 mm) and pelleted (P). Access to the feed was restricted to 4 h.

	Treatment	C	P	SEM	*p*-Value		
	Particle Size (PS; mm)				PS	Treatment	Period
a + b	0.5–1.0	57.0	56.0	1.05	0.0015	0.6036	0.8984
	1.0–2.0	57.2	57.1				
	2.0–4.0	56.2	55.7				
	4.0–8.0	53.7	54.1				
c	0.5–1.0	0.115	0.112	0.0089	0.0051	0.0024	0.0466
	1.0–2.0	0.122	0.080				
	2.0–4.0	0.089	0.076				
	4.0–8.0	0.098	0.071				

SEM: standard error of the mean.

## Data Availability

The original contributions presented in the study are included in the article, further inquiries can be directed to the corresponding author.

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
