# Peer review of "Presentation (Chopped Versus Ground and Pelleted) of a Low-Quality Alfalfa Hay in Sheep: Effects on Intake, Feeding Behaviour, Rumen Fill and Digestion, and Passage"

_animals, 2025, doi:10.3390/ani15040541_

Round 1

Reviewer 1 Report

Comments and Suggestions for Authors

Review:  Alfalfa hay presentation (chopped versus ground and pelleted) in sheep: Effects on intake, feeding behaviour, rumen fill and digestion, and passage  

Antonio de Vega *, Josep Gasa , Carlos Castrillo and José Antonio Guada

General comments

This is a study examining the effects of pelleting on intake of alfalfa hay and the mechanism by which the differences in intake come about. It is a body of work which is well done and uses appropriate methods, with some limitations, but conducted well. It will be a valuable addition to the literature in this field.

There is a large body of work on the effect of pelleting on intake and the effects are well known ie intake increases and digestibility declines. There is nothing new in this aspect of the study. I think the preamble to the study detracts from some of its more important points and I will allude to these in more detail later. However invoking circular agriculture and the effects of pelleting as major points of the study are not very important issues for this study. The authors need to make the point more strongly, and obviously, that they are looking at the mechanism behind why pelleting increases intake. This in itself is not new as Laredo and Minson British Journal of Nutrition (1975) 33, 159-170 and Faichney have determined that reduced retention time is the major factor leading to an increased intake. So why do the study at all and what does it add to the literature? It seems to me that the authors want to examine the behaviour of small particles within the rumen and determine if their retention time by passage or digestion pathways alter under pelleting and control intake. Thus a major contribution of the study is the particle behaviour and the role in intake and the conditions of the rumen mat and other physical parameters of the rumen digesta. Rumination and mastication are both factors which contribute to particle size reduction and passage and it is known that pelleting can affect both and the study has detailed measurements to contribute to this. These are thus very important mechanisms by which pelleting offers a means to study particle movement and it seems to me that the experiment sets out to look at these. A study simply on pelleting alfalfa is not very interesting nor contributes much to what is already known. After all pelleted alfalfa is a well established product in many parts of the world now. So why give us another study into a well established fact and industry. The novelty of the study is the mechanism by which the increase in intake comes about and I think the abstract, aims, introduction etc need to focus on this aspect of the study. The more applied aspects just detract from the paper.

The study has used established marker techniques to examine some of these mechanisms. The discussion etc should emphasize that the application is under non-steady state conditions and irrespective of whether the application of markers has been “rehabilitated” the mathematical basis and assumptions remain the same and without due care to marker kinetics can lead to erroneous conclusions. I think the authors should be very clear about the limitations of marker kinetics under non-steady state conditions. Just the one sentence (L 82) does not give sufficient justification to the use of markers under these conditions and the limitations of such data. They need to highlight this otherwise others less experienced than this group apply marker kinetics without realising the limitations under non-steady state conditions. They have a responsibility to inform future researchers about the use of markers under these conditions. Having said that I do not have a problem with how they applied the marker kinetics to the study and I look at the data understanding the limitations but nevertheless able to assess the differences and their biological significance.

A major finding in the study was the determination of ks, the marker dilution rate that they isolated from the 24 hour plus marker kinetics by curve peeling the marker dilution curve. This is a very important finding but more importantly is the understanding and determination of this value from the overall curve. First I think nearly all previous authors have ignored this concept that an increase in the pool size over a measurement period leads to a dilution curve, part of which is due to loss of marker from the pool and part which is a consequence of an increase in pool size, here due to eating and an increase in rumen volume. I think more should be made of this finding and in the application of marker kinetics. This is a very original finding and application.

I was surprised that there was no measurement of digestibility in the study. I understand the limitations that pens etc can impose but it seems a basic measurement that is needed to highlight the effect that pelleting has on this parameter. The authors have faecal marker curves and the application of these would enable faecal output and hence DM digestibility to be calculated. It would add to the paper. Papers by France et al (1988) J Theoretical Biology 135, 383-391 and Susmel et al (2007) British Journal of Nutrition 75, 521-532 outline the procedures to be used to estimate faecal output and hence digestibility.

The paper outlines how the reduction in particle size and the passage of particles from the rumen evolves over time with implications for what controls intake. More should be made of this aspect of the study as it is not very apparent on first reading the paper given the title and the preamble to the study.

Specific points

L36 ..significant amounts of particles < 0.15 mm, would be the main responsible for.. should be “main factor…”

L47 …animal performance [8], and researches.. Researches should be “researchers” here and throughout it needs to be changed.

L54 Laredo and Minson (British Journal of Nutrition (1975) 33, 159-170) showed no change in rumen fill and a decline in retention time leading to an increase in intake. Need to include.

L85 transit studies. See later comment.

L91 What does this sentence mean? Rewrite. As an example, the relationships between particle size and digestion and passage kinetics are not sufficiently clear as it is not the effect of rumen digesta particle size on marker dilution kinetics in the compartment.

L101 Six 4-year-old rumen-fistulised…  This should be rumen fistulated

L102 .. and average.. Should be …an average…

Table 1 Why does C diet have such a high proportion of particles <0.15mm compared to the P diet. If it was simply leaf shattering and dust one would expect that grinding and pelleting would have as big if not bigger impact. This is a very strange result and worthy of a comment.

L117 The procedure for feeding etc is adequately described but the length of each feeding period which appears to be over 21 days needs to be clearly stated and similarly it is implied that intake was measured ad lib yet nowhere is the procedure described ie what level of refusal etc. These need better descriptions. Or is it simply the intake achieved over 4 hrs?

L117 why feed only over 4 hours? I assume it was to enable the changes in particle size and the marker kinetics to be adequately described and measured given the non-steady state conditions of the experiment. It is worth stating why this procedure was adopted.

L119 …before feeding, for transit studies. This description “transit studies” is incorrect here, earlier and later.  Transit time is used to describe the time taken for marker to move between two points eg rumen to duodenum, ileum or faeces. The study looks at marker kinetics in a mixing pool and the appropriate terms are retention time, fractional passage (or outflow rate) or T1/2 of marker in the pool. Perhaps simply say here and elsewhere “marker kinetic studies”. Transit studies or time should be deleted here and elsewhere except if a description is made of say first appearance of marker in the faeces which you might determine if you apply the France equation to the faecal marker data.

L119 Labelled diet C was previously chopped down with scissors… This seems a strange procedure to simulate mastication during eating and does not really masticate the plant material. Why choose 5mm? Better description needed of what actually done and rationale.

L125 as previously stated you have faecal marker values enabling you to determine faecal output by both markers. Such a comparison would be interesting and similarly the faecal output could be used to estimate digestibility. Suggest you look at this calculation.

L148 what is different to taking whole digesta for CoEDTA? Was there a difference between the two times or methods? Later you mention that the whole digesta was different to the fluid estimate. I’ll elaborate later but I cannot see why the values would be different between dried whole digesta and fluid values or the faecal values. CoEDTA is either a water soluble marker or it is not. There is enough data from a lot of studies to indicate it is a suitable fluid phase marker although some EDTA markers have been shown to have some particle attachment but not at enough levels to alter its fluid phase properties. I think there are other possible reasons why the marker values for outflow rate should differ. See later.

L152 need to specify the length of each period.

L158 Shouldn’t this be amylase?

L168 faces should be “faeces

Figure 1 I think this is one of the most important findings of this work and more emphasis should be made of the theory behind it.

L197 ..liquid-volume decrease. For a period of 24 h, and.. Something is wrong with this sentence. Rewrite.

L212 L,in *is this Lmin?

L241. Would you really expect these animals (class and weight) to increase live weight over 3 weeks and could you detect it. I’d just say there was no weight change and you could specify the weights as you have.

L251 …was expressed as min/kg DM… Here and elsewhere the units are not adequately explained. This also applies to Tables. Check carefully throughout.  Should be : was expressed as min/kg DMI

Table 2 Intake of particles < 0.15 mm (g/d) Why is intake of P particles < 0.15 so much greater than if amount offered much greater in C. See query as to why a difference in initial composition.

Table 3. Units here and elsewhere need careful description as currently not clear.

Rate of reduction. Units?

Proportion should be Percentage

L312 c y p is y is that Spanish for  c vs p?

Figure 3 These are Percentages not proportions. Change.

L318 Figure 3 Access not accession.

Table 4. Need units for L, WD, DM etc

L325 Table 5 need units for kr etc as in Table 4.

L327 Table 5 use a different term kf to differentiate to above in Table 4 eg kfa

L333 …animals consumed a 34 % more feed… Delete “a” and simply “animals consumed 34 % more feed”

L335 I am assuming you mean that intake was limited by rumen fill which was reached in a short period of time for animals fed the P diet? Why not say that. Also you should reference  Laredo and Minson (1975) paper referenced earlier for rumen fill effects of pelleted diets and they were done under steady state conditions.

L354 To this respect.. should be “In this respect…”

L375 Proposing that CoEDTA could be a marker for the solid phase also is not correct based on your later assumptions for CoEDTA in fluid phase. This section is confusing. For CoEDTA in WD and faeces, isn't the CoEDTA still in the water and so is still  a water marker. If no difference does this mean water and particles have same RT or FOR. This seems wrong. What was the difference between CoEDTA measured in WD and in liquid as two injections were done.  Where are data? Isn’t a probable explanation that the DM content varies with the sampling times and by analysing the samples in dried material and so expressing the values/unit DM going to influence your result. Even if some CoEDTA binds to particles (<10% in CrEDTA studies) this would not give you the data that you show. I think a few other possible explanations are needed for this outcome.

L377 “The lack of differences in passage rate between diets, regardless the marker employed, contradicts the long established idea of a faster rate for ground and pelleted diets [30,52,53]. Again, the faster dilution rate of the liquid phase with diet P, likely including significant amounts of particles < 0.15 mm, would indicate a faster passage of this diet”. Table 5 shows faster k value for P than C but WD data and faecal not the same. Why? You have one set of data showing a difference and one with no difference. Which do you believe and why? This is not in your conclusions nor abstract.

L406 should be: These values…

L417 Differences in potential degradability might have been significant but not great in quantitative terms and within the sort of variation one gets from plotting degradation curves. Perhaps alluding to the small differences rather than simply the significant differences would be better.

L435 …would be the main responsible. Should be: ..would be the main factor responsible.

Author Response

Dear Reviewer,

Thank you very much for your comments and suggestions which undoubtedly will improve our paper. Please find below a detailed answer to all your comments

General comments:

This is a study examining the effects of pelleting on intake of alfalfa hay and the mechanism by which the differences in intake come about. It is a body of work which is well done and uses appropriate methods, with some limitations, but conducted well. It will be a valuable addition to the literature in this field.

There is a large body of work on the effect of pelleting on intake and the effects are well known ie intake increases and digestibility declines. There is nothing new in this aspect of the study. I think the preamble to the study detracts from some of its more important points and I will allude to these in more detail later. However, invoking circular agriculture and the effects of pelleting as major points of the study are not very important issues for this study. The authors need to make the point more strongly, and obviously, that they are looking at the mechanism behind why pelleting increases intake. This in itself is not new as Laredo and Minson British Journal of Nutrition (1975) 33, 159-170 and Faichney have determined that reduced retention time is the major factor leading to an increased intake. So why do the study at all and what does it add to the literature? It seems to me that the authors want to examine the behaviour of small particles within the rumen and determine if their retention time by passage or digestion pathways alter under pelleting and control intake. Thus a major contribution of the study is the particle behaviour and the role in intake and the conditions of the rumen mat and other physical parameters of the rumen digesta. Rumination and mastication are both factors which contribute to particle size reduction and passage and it is known that pelleting can affect both and the study has detailed measurements to contribute to this. These are thus very important mechanisms by which pelleting offers a means to study particle movement and it seems to me that the experiment sets out to look at these. A study simply on pelleting alfalfa is not very interesting nor contributes much to what is already known. After all, pelleted alfalfa is a well-established product in many parts of the world now. So why give us another study into a well-established fact and industry. The novelty of the study is the mechanism by which the increase in intake comes about and I think the abstract, aims, introduction etc. need to focus on this aspect of the study. The more applied aspects just detract from the paper.

 Response: The introduction and aims of the paper have been amended according to your suggestions (L48-52, L72-78 and L81-84), focusing now on the mechanisms by which the increase in intake due to grinding and pelleting comes about. The paragraph invoking circular agriculture has been deleted, as well as the more applied aspects of the paper.

 The study has used established marker techniques to examine some of these mechanisms. The discussion etc should emphasize that the application is under non-steady state conditions and irrespective of whether the application of markers has been “rehabilitated” the mathematical basis and assumptions remain the same and without due care to marker kinetics can lead to erroneous conclusions. I think the authors should be very clear about the limitations of marker kinetics under non-steady state conditions. Just the one sentence (L 82) does not give sufficient justification to the use of markers under these conditions and the limitations of such data. They need to highlight this otherwise others less experienced than this group apply marker kinetics without realising the limitations under non-steady state conditions. They have a responsibility to inform future researchers about the use of markers under these conditions. Having said that I do not have a problem with how they applied the marker kinetics to the study and I look at the data understanding the limitations but nevertheless able to assess the differences and their biological significance.

 Response: The non-steady state conditions of our experiment have been highlighted (L107, L180-186), and the limitations derived from that fact pointed out (L180-186).

A major finding in the study was the determination of ks, the marker dilution rate that they isolated from the 24 hour plus marker kinetics by curve peeling the marker dilution curve. This is a very important finding but more importantly is the understanding and determination of this value from the overall curve. First I think nearly all previous authors have ignored this concept that an increase in the pool size over a measurement period leads to a dilution curve, part of which is due to loss of marker from the pool and part which is a consequence of an increase in pool size, here due to eating and an increase in rumen volume. I think more should be made of this finding and in the application of marker kinetics. This is a very original finding and application.

 Response: some discussion has been added about the meaning of kf and ks (L379-391). We are aware that this discussion is limited, and look forward to your suggestions to improve it.

 I was surprised that there was no measurement of digestibility in the study. I understand the limitations that pens etc can impose but it seems a basic measurement that is needed to highlight the effect that pelleting has on this parameter. The authors have faecal marker curves and the application of these would enable faecal output and hence DM digestibility to be calculated. It would add to the paper. Papers by France et al (1988) J Theoretical Biology 135, 383-391 and Susmel et al (2007) British Journal of Nutrition 75, 521-532 outline the procedures to be used to estimate faecal output and hence digestibility.

 Response: faecal output values have been estimated from faecal Yb and Co excretion curves, and together with intake used to estimate digestibility.

The paper outlines how the reduction in particle size and the passage of particles from the rumen evolves over time with implications for what controls intake. More should be made of this aspect of the study as it is not very apparent on first reading the paper given the title and the preamble to the study.

 Response: the introduction has been changed to outline how the reduction in particle size and the passage of particles from the rumen evolves over time, and their implication in the control of intake.

Specific points

L36 ..significant amounts of particles < 0.15 mm, would be the main responsible for.. should be “main factor…”

Response: the sentence has been amended according to your suggestions (L38), and the changes highlighted in yellow in the new version of the paper.

L47 …animal performance [8], and researches.. Researches should be “researchers” here and throughout it needs to be changed.

Response: ‘researches’ has been substituted with ‘researchers’ (L42).

L54 Laredo and Minson (British Journal of Nutrition (1975) 33, 159-170) showed no change in rumen fill and a decline in retention time leading to an increase in intake. Need to include.

Response: a sentence pointing out that the main reason for the higher voluntary intake of pelleted vs. chopped low-quality forages seems to be a reduced retention time of digesta rather than changes in rumen fill. The paper by Laredo and Minson (1975) has been cited (L48-52, L356-358, and L534-535).

L85 transit studies. See later comment.

Response: transit studies has been replaced with marker kinetic studies where appropriate.

L91 What does this sentence mean? Rewrite. As an example, the relationships between particle size and digestion and passage kinetics are not sufficiently clear as it is not the effect of rumen digesta particle size on marker dilution kinetics in the compartment.

Response: this sentence has been deleted.

L101 Six 4-year-old rumen-fistulised…  This should be rumen fistulated…

Response: this has been changed.

L102 .. and average.. Should be …an average…

Response: this has been changed.

Table 1 Why does C diet have such a high proportion of particles <0.15mm compared to the P diet. If it was simply leaf shattering and dust one would expect that grinding and pelleting would have as big if not bigger impact. This is a very strange result and worthy of a comment.

Response: sorry for the mistake, but the values for the <0.15mm fraction were not correct. Table 1 has been amended with the corrections highlighted in yellow.

L117 The procedure for feeding etc is adequately described but the length of each feeding period which appears to be over 21 days needs to be clearly stated and similarly it is implied that intake was measured ad lib yet nowhere is the procedure described ie what level of refusal etc. These need better descriptions. Or is it simply the intake achieved over 4 hrs?

Response: each period of the cross over lasted 25 days (15 for adaptation to the diets, 4 for obtaining faecal marker excretion curves, 1 for rumen sampling to determine the amounts of different particle sizes over a 24 h period, 3 for in situ rumen degradation, 1 for marker dilution rate in the rumen, and 1 for feeding behaviour). All this has been stated in L107-113, L116, L128, L131, L143 and L151. Feed was offered to allow 20% refusals (L103)

L117 why feed only over 4 hours? I assume it was to enable the changes in particle size and the marker kinetics to be adequately described and measured given the non-steady state conditions of the experiment. It is worth stating why this procedure was adopted.

Response: Yes, the restriction was imposed to allow for the study of rumen digesta kinetics, and of the effects of intake and chewing during eating and ruminating on rumen fill and digesta degradation. This has been commented in L104-107.

L119 …before feeding, for transit studies. This description “transit studies” is incorrect here, earlier and later.  Transit time is used to describe the time taken for marker to move between two points eg rumen to duodenum, ileum or faeces. The study looks at marker kinetics in a mixing pool and the appropriate terms are retention time, fractional passage (or outflow rate) or T1/2 of marker in the pool. Perhaps simply say here and elsewhere “marker kinetic studies”. Transit studies or time should be deleted here and elsewhere except if a description is made of say first appearance of marker in the faeces which you might determine if you apply the France equation to the faecal marker data.

Response: marker kinetic studies has been used where appropriate.

L119 Labelled diet C was previously chopped down with scissors… This seems a strange procedure to simulate mastication during eating and does not really masticate the plant material. Why choose 5mm? Better description needed of what actually done and rationale.

Response: diet C was not chopped down to simulate mastication but to facilitate the introduction of the labelled material into the rumen through the cannula. This has been done in previous work by our group (https://doi.org/10.1017/S0007114598001445, https://doi.org/10.1071/AR97119). The sentence has been changed (L119-120).

L125 as previously stated you have faecal marker values enabling you to determine faecal output by both markers. Such a comparison would be interesting and similarly the faecal output could be used to estimate digestibility. Suggest you look at this calculation.

Response: faecal output has been estimated from Co and Yb faecal excretion curves (L186-188) with no differences between markers (L256-258). The results have been discussed in L358-369.

L148 what is different to taking whole digesta for CoEDTA? Was there a difference between the two times or methods? Later you mention that the whole digesta was different to the fluid estimate. I’ll elaborate later but I cannot see why the values would be different between dried whole digesta and fluid values or the faecal values. CoEDTA is either a water soluble marker or it is not. There is enough data from a lot of studies to indicate it is a suitable fluid phase marker although some EDTA markers have been shown to have some particle attachment but not at enough levels to alter its fluid phase properties. I think there are other possible reasons why the marker values for outflow rate should differ. See later.

Response: yes, the sampling times were different (see L122-124, L145 and L183-186), and that was probably the reason for the discrepancies between the results offered in Tables 4 and 5. The discussion has been changed (L411-422).

L152 need to specify the length of each period.

Response: already dealt with above.

L158 Shouldn’t this be amylase?

Response: this has been changed (L161).

L168 faces should be “faeces”

Response: this has been changed (L171).

Figure 1 I think this is one of the most important findings of this work and more emphasis should be made of the theory behind it.

Response: please see the discussion in L379-391.

L197 ..liquid-volume decrease. For a period of 24 h, and.. Something is wrong with this sentence. Rewrite.

Response: the sentence has been rewritten (L209-213).

L212 L,in *is this Lmin?

Response: sorry for the typographical error. It has been corrected (L223).

L241. Would you really expect these animals (class and weight) to increase live weight over 3 weeks and could you detect it. I’d just say there was no weight change and you could specify the weights as you have.

Response: as there was no weight change, this sentence has been deleted.

L251 …was expressed as min/kg DM… Here and elsewhere the units are not adequately explained. This also applies to Tables. Check carefully throughout.  Should be : was expressed as min/kg DMI

Response: sorry for the carelessness. Units have been checked throughout (L264 and Table 3).

Table 2 Intake of particles < 0.15 mm (g/d) Why is intake of P particles < 0.15 so much greater than if amount offered much greater in C. See query as to why a difference in initial composition.

Response: as pointed out above, there was a mistake and the values for the <0.15mm fraction were not correct. They have been corrected now.

Table 3. Units here and elsewhere need careful description as currently not clear.

Rate of reduction. Units?

Proportion should be Percentage

Response: Units have been checked.

L312 c y p is y is that Spanish for  c vs p?

Response: sorry for the mistake.’y’ has been substituted with ‘vs’ (L345).

Figure 3 These are Percentages not proportions. Change.

Response: this has been changed.

L318 Figure 3 Access not accession.

Response: this has been changed.

Table 4. Need units for L, WD, DM etc

Response: units are in the table head: litres for Lmin and Lmax, and grams for WDmin, WDmax, DMmin and DMmax.

L325 Table 5 need units for kr etc as in Table 4.

Response: units (h-1) have been included.

L327 Table 5 use a different term kf to differentiate to above in Table 4 eg kfa

Response: kf has been substituted with kfa.

L333 …animals consumed a 34 % more feed… Delete “a” and simply “animals consumed 34 % more feed”

Response: this has been changed (L355).

L335 I am assuming you mean that intake was limited by rumen fill which was reached in a short period of time for animals fed the P diet? Why not say that. Also you should reference  Laredo and Minson (1975) paper referenced earlier for rumen fill effects of pelleted diets and they were done under steady state conditions.

Response: the sentence has been rewritten, and the reference to Laredo and Minson (1975) included (L356-358)

L354 To this respect.. should be “In this respect…”

Response: this has been changed (L395).

L375 Proposing that CoEDTA could be a marker for the solid phase also is not correct based on your later assumptions for CoEDTA in fluid phase. This section is confusing. For CoEDTA in WD and faeces, isn't the CoEDTA still in the water and so is still  a water marker. If no difference does this mean water and particles have same RT or FOR. This seems wrong. What was the difference between CoEDTA measured in WD and in liquid as two injections were done.  Where are data? Isn’t a probable explanation that the DM content varies with the sampling times and by analysing the samples in dried material and so expressing the values/unit DM going to influence your result. Even if some CoEDTA binds to particles (<10% in CrEDTA studies) this would not give you the data that you show. I think a few other possible explanations are needed for this outcome.

Response: as pointed out above, the discussion regarding this issue has been changed (L411-422).

L377 “The lack of differences in passage rate between diets, regardless the marker employed, contradicts the long established idea of a faster rate for ground and pelleted diets [30,52,53]. Again, the faster dilution rate of the liquid phase with diet P, likely including significant amounts of particles < 0.15 mm, would indicate a faster passage of this diet”. Table 5 shows faster k value for P than C but WD data and faecal not the same. Why? You have one set of data showing a difference and one with no difference. Which do you believe and why? This is not in your conclusions nor abstract.

Response: please see the answer to the previous comment.

L406 should be: These values…

Response: this has been changed (L447).

L417 Differences in potential degradability might have been significant but not great in quantitative terms and within the sort of variation one gets from plotting degradation curves. Perhaps alluding to the small differences rather than simply the significant differences would be better.

Response: the sentence has been changed according to your suggestion (L458-460).

L435 …would be the main responsible. Should be: ..would be the main factor responsible.

Response: this has been changed (L476).

Kind regards,

Antonio de Vega

Reviewer 2 Report

Comments and Suggestions for Authors

The authors evaluated two forms of alfalfa hay presentation for sheep, differing in particle size. The study was well conducted and employed liquid phase markers, Co and Yb, to assess the dilution rate on the particles in the rumen. Additionally, the ingestive behavior of sheep fed these two diets was analyzed. However, several concerns arise due to the simplicity and lack of clarity in the description of the material and methods. The information provided is overly concise and, at times, confusing. There are also uncertainties regarding the study's hypotheses. It seems the authors also aimed to identify the best marker to estimate the flow of the liquid phase, but this objective is not explicitly stated. Another important aspect is the absence of data on diet intake and digestibility. The results presented are insufficient to draw meaningful conclusions about the impact of alfalfa hay presentation on sheep performance. Furthermore, there is no discussion on whether the pellet preparation is viable, despite its apparent effect on reducing feeding time. A stronger contextualization of the study and its practical applications is needed. Even for a theoretical study, including data on the impact of these treatments on organic matter digestibility, rumen pH, or microbial production would enhance the relevance of the findings. The current focus on liquid phase kinetics and feeding behavior seems too narrow, offering only limited insights into the kinetics of fiber digestion in sheep. Despite these limitations, the study could be significantly improved with better organization and presentation. The materials and methods section requires more detail and chronological structure. Each stage of the study should have its objective clearly stated, and the hypotheses being tested should be explicitly defined. Other detailed comments are place bellow.

Simple summary: There is a conceptual issue in this section, as well as in other parts of the text, including the introduction. Alfalfa hay should not be described as a low-quality forage; on the contrary, it is a high-quality feed, characterized by high digestibility and a rich content of soluble protein. Low-quality forages, on the other hand, are typically C4 grasses, commonly grown in tropical environments. Therefore, the text should be adjusted to avoid creating the expectation that this study involves low-quality forages. Instead, it should remain focused on the actual object of study, which is how the form of presentation of alfalfa affects animal performance.

Abstract: Half of the abstract is taken up by an introduction that closely resembles the content of the previous section (simple summary). In this section, it is essential to focus on the hypothesis of the study, the detailed objectives, as well as the main results and conclusions. Please revise the abstract to ensure it succinctly presents the core elements of the work, emphasizing its originality and key findings.

Introduction: The first paragraph refers to low-quality forage, which does not accurately reflect the scope of the study, as previously mentioned. Please revise this section to ensure it aligns with the actual focus of the work. The third paragraph focuses solely on the characteristics and idiosyncrasies of fiber for ruminants. This discussion is unnecessary, as it presents information that is already well-known and obvious. Additionally, it does not contribute to the understanding of the study and unnecessarily lengthens the introduction. For these reasons, this paragraph could be removed. Please add the hypothesis of the work before the objectives.

Material and methods: 

In Table 1, Why are the diets presented as different? The bulk remains the same, with only the particle sizes of the diet varying. Was concentrate included in the diets? If so, what is the ratio between bulk and concentrate? Please clarify these aspects to provide a clearer understanding of the experimental design. How was the intake measured? Composite samples were made from which days?

Line 116 “Feed was offered once a day (9:00), with access restricted to 4 h.” Why was this time restriction applied? How many periods were carried out? How was Co-EDTA applied? With a pump or manually? Please make a more detailed description of the experimental periods and markers application. The sentence is huge confuse.

Line 131: How was made the particle size intake and fecal distribution? 

Line 132: Why were these different particle sizes incubated? What was the objective of this procedure? On which day of the experimental period was this protocol applied? Please include detailed information about the experimental procedures in a clear chronological order. For each procedure, it is essential to specify its objective to provide better clarity. As currently presented, this section is confusing and does not allow for replication of the trial based solely on the material and methods described.

Line 151 and 168: Please describe the procedures.

Results:

Table 3: How were these rates calculated? Please add in material and methods.

Figure 3: was there any statistical difference for the different particle sizes inside each time? Please add the stats information in the figures.

Table 4: I could not see how experimental periods would be important in this analysis. The experimental design was not clear in the material and methods.

Table 5: Why were the authors expecting alteration in the dilution rate from different markers? Which type of test they would apply? Which was the hypothesis associated with this. Too many concerns about methodology and hypothesis were not clear.

The Discussion and Conclusion sections should emphasize the contributions of this work to understanding the effects of different particle sizes on sheep. Specifically, the authors should address the impact of grinding and pelleting feed. Do these processes provide a practical benefit for sheep performance? Are there improvements in digestibility, feed intake, or overall animal performance? If the current data are insufficient to draw firm conclusions, it is important to outline the types of additional studies required to reach these conclusions. For example, further research could explore the effects of feed processing on broader parameters such as nutrient digestibility, rumen fermentation, microbial populations, and overall performance. I am not sure whether these degradability data performed during ruminal collection processes can be reliable. Furthermore, it is not clear in what experimental period the degradability procedures were performed. Additionally, the authors must acknowledge the limitations of their study. Since the study primarily evaluates the dilution rate of particles and feeding behavior, the data are limited. The authors should explicitly state these limitations and suggest future research directions to make their findings more applicable and informative. This will help expand the relevance of their work and guide future studies in this area.

Author Response

Dear Reviewer,

Thank you very much for your comments and suggestions which undoubtedly will improve our paper. Please find below a detailed answer to all your comments

The authors evaluated two forms of alfalfa hay presentation for sheep, differing in particle size. The study was well conducted and employed liquid phase markers, Co and Yb, to assess the dilution rate on the particles in the rumen. Additionally, the ingestive behavior of sheep fed these two diets was analyzed. However, several concerns arise due to the simplicity and lack of clarity in the description of the material and methods. The information provided is overly concise and, at times, confusing. There are also uncertainties regarding the study's hypotheses. It seems the authors also aimed to identify the best marker to estimate the flow of the liquid phase, but this objective is not explicitly stated. Another important aspect is the absence of data on diet intake and digestibility. The results presented are insufficient to draw meaningful conclusions about the impact of alfalfa hay presentation on sheep performance. Furthermore, there is no discussion on whether the pellet preparation is viable, despite its apparent effect on reducing feeding time. A stronger contextualization of the study and its practical applications is needed. Even for a theoretical study, including data on the impact of these treatments on organic matter digestibility, rumen pH, or microbial production would enhance the relevance of the findings. The current focus on liquid phase kinetics and feeding behavior seems too narrow, offering only limited insights into the kinetics of fiber digestion in sheep. Despite these limitations, the study could be significantly improved with better organization and presentation. The materials and methods section requires more detail and chronological structure. Each stage of the study should have its objective clearly stated, and the hypotheses being tested should be explicitly defined. Other detailed comments are place bellow.

Response: The paper has been extensively modified regarding the study’s hypothesis, the description of the materials and methods used, the presentation of the results, the discussion and the conclusion. We are aware that maybe more will be needed to be done, and look forward to receiving your future comments. Rumen fermentation or microbial production were not assessed as they were not the objective of the study. It is worth mentioning that Molaei et al. (2021; Animal Feed Science and Technology 279, 115031; https://doi.org/10.1016/j.anifeedsci.2021.115031) did not find any difference in rumen fermentation variables or microbial production between chopped vs. pelleted alfalfa hay in dairy calves.

Simple summary: There is a conceptual issue in this section, as well as in other parts of the text, including the introduction. Alfalfa hay should not be described as a low-quality forage; on the contrary, it is a high-quality feed, characterized by high digestibility and a rich content of soluble protein. Low-quality forages, on the other hand, are typically C4 grasses, commonly grown in tropical environments. Therefore, the text should be adjusted to avoid creating the expectation that this study involves low-quality forages. Instead, it should remain focused on the actual object of study, which is how the form of presentation of alfalfa affects animal performance.

Response: the alfalfa hay used in the present experiment was of low digestibility (see Table 2) and that is why we considered it a low-quality alfalfa hay. The alfalfa hay produced in our area is usually of low digestibility (please see https://doi.org/10.1017/S0007114598001445 and https://doi.org/10.1071/AR97119). We have pointed out this issue throughout the text (highlighted in yellow).

Abstract: Half of the abstract is taken up by an introduction that closely resembles the content of the previous section (simple summary). In this section, it is essential to focus on the hypothesis of the study, the detailed objectives, as well as the main results and conclusions. Please revise the abstract to ensure it succinctly presents the core elements of the work, emphasizing its originality and key findings.

Response: the abstract has been rewritten trying to follow your suggestions.

Introduction: The first paragraph refers to low-quality forage, which does not accurately reflect the scope of the study, as previously mentioned. Please revise this section to ensure it aligns with the actual focus of the work. The third paragraph focuses solely on the characteristics and idiosyncrasies of fiber for ruminants. This discussion is unnecessary, as it presents information that is already well-known and obvious. Additionally, it does not contribute to the understanding of the study and unnecessarily lengthens the introduction. For these reasons, this paragraph could be removed. Please add the hypothesis of the work before the objectives.

Response: please see our answer to your comment above regarding the use of a low-quality alfalfa hay in the present experiment. The paragraph dealing with fibre characteristics has been removed, and the hypothesis of the work included (L81-84).

Material and methods: 

In Table 1, Why are the diets presented as different? The bulk remains the same, with only the particle sizes of the diet varying. Was concentrate included in the diets? If so, what is the ratio between bulk and concentrate? Please clarify these aspects to provide a clearer understanding of the experimental design. How was the intake measured? Composite samples were made from which days?

Response: the pelleted diet had only a 3% lower content in organic matter, and an 8% lower content in crude protein compared to the chopped diet. This could have been due to leaf shattering and dust during the pelleting process. On the other hand, and probably due to the same leaf losses, fibre content was higher in the pelleted diet (6% more NDF and 16% more lignin, with no differences in ADF). We apologise the NDF, ADF and lignin contents of the two diets were switched in the original version of the paper. They have been corrected now. Concentrate was not included in the diet (please see L103-104). Information about how intake was measured has been included (L108-113).

Line 116 “Feed was offered once a day (9:00), with access restricted to 4 h.” Why was this time restriction applied? How many periods were carried out? How was Co-EDTA applied? With a pump or manually? Please make a more detailed description of the experimental periods and markers application. The sentence is huge confuse.

Response: the restriction in the access to the feed was imposed to allow for the study of rumen digesta kinetics, and of the effects of intake and chewing during eating and ruminating on rumen fill and digesta degradation, in non-steady state conditions. This has been added in L104-107 of the revised version of the paper. The design was a cross-over with two diets (L92-94), hence two experimental periods. Co-EDTA was added manually with the aid of a 50 ml syringe (L118-119). Experimental periods have now been described in more detail (L116-155).

Line 131: How was made the particle size intake and fecal distribution? 

Response: knowing the dry matter intake (Table 2), and the particle size distribution (Table 1), intake of the different particles was easily calculated. Particle size distribution of faeces was measured as suggested in https://doi.org/10.1017/S0007114598001445 (L171).

Line 132: Why were these different particle sizes incubated? What was the objective of this procedure? On which day of the experimental period was this protocol applied? Please include detailed information about the experimental procedures in a clear chronological order. For each procedure, it is essential to specify its objective to provide better clarity. As currently presented, this section is confusing and does not allow for replication of the trial based solely on the material and methods described.

Response: the objective of this procedure was to assess the influence of the rumen environment set by the form of presentation of the alfalfa hay on rumen degradation of different particle sizes (L133-135). This was done on day 21 of each experimental period of the cross-over (L131).

Line 151 and 168: Please describe the procedures.

Response: visual observations of feeding behaviour were made once every 3 min on each animal, and three categories were used to classify the animals’ behaviour: chewing during eating, ruminating and idling (L152-154). Particle size distribution of faeces and rumen content was determined using a wet-sieving apparatus as previously described in https://doi.org/10.1017/S0007114598001445 (L171).

Results:

Table 3: How were these rates calculated? Please add in material and methods.

Response: this has been described in L227-L240.

Figure 3: was there any statistical difference for the different particle sizes inside each time? Please add the stats information in the figures.

Response: Dear reviewer, the statistical analysis produced a lengthy table which results we were not able to include in Figure 3. I attach the table and look forward to knowing your decision about whether it is worthy or not to include these results in the paper. The values for C and P are g/100 g DM.

Sampling time

Particle size

Diet C

Diet P

SEM

p-value

H0

>2.4 mm

4.05

1.7

0.978

0.0524

1.2-2.4 mm

9.84

1.72

0.461

<0.0001

0.6-1.2 mm

22.29

21.47

2.849

0.7842

0.3-0.6 mm

17.02

22.09

1.051

0.0029

0.15-0.3 mm

11.28

14.13

1.192

0.0539*

<0.15 mm

35.52

38.90

2.713

0.2600

H4

>2.4 mm

7.64

1.25

0.805

0.0002

1.2-2.4 mm

12.82

2.06

0.304

<0.0001

0.6-1.2 mm

17.10

17.85

1.451

0.6218

0.3-0.6 mm

12.15

17.80

0.551

<0.0001

0.15-0.3 mm

9.76

15.50

0.389

<0.0001

<0.15 mm

40.54

45.49

2.541

0.0991

H8

>2.4 mm

8.08

1.16

0.640

<0.0001

1.2-2.4 mm

13.62

2.00

0.252

<0.0001

0.6-1.2 mm

17.07

19.31

1.288

0.1326

0.3-0.6 mm

11.57

17.30

0.455

<0.0001

0.15-0.3 mm

8.29

14.61

0.564

<0.0001

<0.15 mm

41.36

45.62

1.945

0.0709

H12

>2.4 mm

7.02

1.83

0.977

0.0018

1.2-2.4 mm

13.61

2.40

0.369

<0.0001

0.6-1.2 mm

18.90

21.60

1.390

0.0996

0.3-0.6 mm

12.34

17.14

0.373

<0.0001

0.15-0.3 mm

8.47

13.24

0.488

<0.0001

<0.15 mm

39.66

43.79

2.178

0.1066

H16

>2.4 mm

5.74

1.48

0.9355

0.0039

1.2-2.4 mm

10.77

2.42

0.823

<0.0001

0.6-1.2 mm

19.33

20.85

1.490

0.3471

0.3-0.6 mm

12.81

17.93

1.056

0.0029

0.15-0.3 mm

9.11

12.8

0.8934

0.0061

<0.15 mm

42.25

42.85

1.915

0.7652

H20

>2.4 mm

5.81

3.02

2.399

0.3290

1.2-2.4 mm

11.55

3.08

1.049

0.0040

0.6-1.2 mm

19.81

21.98

3.199

0.5453

0.3-0.6 mm

13.10

21.35

1.535

0.0126

0.15-0.3 mm

9.43

13.74

2.151

0.1387

<0.15 mm

40.32

36.84

2.643

0.2791

H24

>2.4 mm

5.17

1.08

0.676

0.0009

1.2-2.4 mm

8.30

2.31

0.598

<0.0001

0.6-1.2 mm

21.55

22.04

3.409

0.8908

0.3-0.6 mm

16.90

20.22

0.711

0.0035

0.15-0.3 mm

12.28

14.03

1.114

0.1658

<0.15 mm

35.8

40.32

2.786

0.1555

Table 4: I could not see how experimental periods would be important in this analysis. The experimental design was not clear in the material and methods.

Response: animals received alfalfa hay either chopped or ground and pelleted, in a cross-over design. This has been clarified in L92-94.

Table 5: Why were the authors expecting alteration in the dilution rate from different markers? Which type of test they would apply? Which was the hypothesis associated with this. Too many concerns about methodology and hypothesis were not clear.

Response: all these concerns have been dealt with in L116-124, L174-178, and L411-422.

The Discussion and Conclusion sections should emphasize the contributions of this work to understanding the effects of different particle sizes on sheep. Specifically, the authors should address the impact of grinding and pelleting feed. Do these processes provide a practical benefit for sheep performance? Are there improvements in digestibility, feed intake, or overall animal performance? If the current data are insufficient to draw firm conclusions, it is important to outline the types of additional studies required to reach these conclusions. For example, further research could explore the effects of feed processing on broader parameters such as nutrient digestibility, rumen fermentation, microbial populations, and overall performance. I am not sure whether these degradability data performed during ruminal collection processes can be reliable. Furthermore, it is not clear in what experimental period the degradability procedures were performed. Additionally, the authors must acknowledge the limitations of their study. Since the study primarily evaluates the dilution rate of particles and feeding behavior, the data are limited. The authors should explicitly state these limitations and suggest future research directions to make their findings more applicable and informative. This will help expand the relevance of their work and guide future studies in this area.

Response: the discussion has been partly rewritten to try and deal with the referee’s concerns.

Kind regards,

Antonio de Vega

Round 2

Reviewer 2 Report

Comments and Suggestions for Authors

The manuscript has been improved a lot, and now is suitable for publication.